# Linking Gut Microbiome and Lipid Metabolism: Moving beyond Associations

**DOI:** 10.3390/metabo11010055

**Published:** 2021-01-15

**Authors:** Santosh Lamichhane, Partho Sen, Marina Amaral Alves, Henrique C. Ribeiro, Peppi Raunioniemi, Tuulia Hyötyläinen, Matej Orešič

**Affiliations:** 1Turku Bioscience Centre, University of Turku and Abo Akademi University, FI-20520 Turku, Finland; partho.sen@utu.fi (P.S.); marina.amaral@utu.fi (M.A.A.); henrique.carachoribeiro@utu.fi (H.C.R.); pemarau@utu.fi (P.R.); matej.oresic@utu.fi (M.O.); 2School of Medical Sciences, Orebro University, 702 81 Orebro, Sweden; 3Department of Chemistry, Orebro University, 702 81 Orebro, Sweden; Tuulia.Hyotylainen@oru.se; 4Oil Crops Research Institute, Chinese Academy of Agricultural Sciences, Wuhan 430062, China

**Keywords:** microbiome, lipidomics, metabolomics, gut, lipids

## Abstract

Various studies aiming to elucidate the role of the gut microbiome-metabolome co-axis in health and disease have primarily focused on water-soluble polar metabolites, whilst non-polar microbial lipids have received less attention. The concept of microbiota-dependent lipid biotransformation is over a century old. However, only recently, several studies have shown how microbial lipids alter intestinal and circulating lipid concentrations in the host, thus impacting human lipid homeostasis. There is emerging evidence that gut microbial communities play a particularly significant role in the regulation of host cholesterol and sphingolipid homeostasis. Here, we review and discuss recent research focusing on microbe-host-lipid co-metabolism. We also discuss the interplay of human gut microbiota and molecular lipids entering host systemic circulation, and its role in health and disease.

## 1. Lipids and Gut Microbes

Lipids are the major structural constituents of cell membranes. As energy storage molecules, they also store almost twice the energy as that liberated from protein or carbohydrate catabolism. Moreover, lipids regulate many essential biological functions, including intra-cellular signaling processes. For instance, sphingolipids (SPs), particularly ceramides, have a part to play in regulation of cell signaling and apoptosis [1]. Other lipids such as diacylglycerols (DGs) act as intermediates of energy metabolism and as signaling molecules [2]. Overall, lipid metabolism exhibits spatial and dynamic complexity at multiple levels. Thus, it is not surprising that lipid disturbances have important physiological consequences impacting human health [3].

The human gut, on the other hand, harbors metabolically-active microbial communities, which have profound impact on the absorption, digestion, metabolism and excretion of lipids [4]. There is a growing consensus that gut microbes and their metabolic activity alter the metabolic state of the host. Most recent studies in this field suggest the integration of the gut microbiome and metabolome, rather than sole focus on microbial taxonomic profiling, affords better understanding of microbiome-mediated (patho)physiological processes [5,6]. Thus, metabolome-based strategies for the study of gut microbial communities at both structural and functional levels are gaining much-warranted increasing attention.

Previous studies of the gut microbiome-metabolome co-axis have mainly focused on water-soluble, polar metabolites (e.g., tryptophan catabolites, amino acids, tricarboxylic acid cycle (TCA) intermediates), while the microbial-host-lipid co-axis has not received as much attention. In this review, we highlight recent fecal lipidomics studies aimed at deciphering the molecular lipids that are secreted, hydrolyzed or transformed by gut microbial communities. We also discuss mechanistic studies, involving integration of shotgun metagenomics and lipidomics data, and how these approaches may lead to better understanding of the impact of microbial lipids on host physiology. To this end, we highlight the emerging applications of genome-scale metabolic modeling (GSMM) as a way to study co-metabolism between the microbes as well as host–microbe interactions via lipidomes.

## 2. Lipid Pool of the Human Gut

A link between host molecular lipids, gut microbiota and human health is already evident, given the associations between them in several clinical studies [7,8]. Gut microbiota can both modulate the amount of energy that is extracted from food during digestion, and synthesize lipids and metabolites that may have an impact on human health. Alterations in the gut microbiota—host lipidome have been linked obesity and the development of obesity-related illnesses, among others. In humans, the diversity of the gut microbiome has been found to have a negative association with BMI and serum triacylglycerols (TGs) [9]. Liu et al. reported that gestational diabetes mellitus comorbidity was strongly associated with both a specific gut microbial composition and with the circulating lipidome [10]. Here, relative abundances of *Faecalibacterium* and *Prevotella* showed linkage with circulating lipids, in particular with lysophosphatidylethanolamine, and phosphatidylglycerols. Recently, Benítez–Páez et al. used a multi-omics approach (metabolomics, lipidomics and shotgun metagenomics) to characterize the impact of arabinoxylan oligosaccharides in overweight individuals [11]. They found that an arabinoxylan-enriched diet altered the host metabolic state, including ceramide and choline levels, which subsequently affected the abundance of *Prevotella* and *Clostridial* species in the gut. Animal studies also posit that gut microbes can modulate the host’s lipidome. For instance, Albouery et al. investigated how colonization of germ-free (GF) mice by the fecal microbiota of young or old donor mice impacted the lipid content of the brain and liver [12]. Mice receiving fecal bacteria from aged mice exhibited increased total monounsaturated fatty acids, and a reduction in the relative amounts of cholesterol and total polyunsaturated fatty acids in the brain cortex. In addition, the transfer of microbiota from aged to young mice modified the relative abundances of the different lipid classes and the fatty acid content of the liver. In another study, Just et al. studied how diets enriched with primary bile acids (BAs) with or without addition of lard or palm oil, impacted gut microbiota composition and function in mice [13]. The authors reported that the lard + BA-enriched diet increased the fat mass of colonized mice, but not in the GF mice, as compared to palm oil. Subsequently, these effects were associated with impaired glucose tolerance and elevated TGs, cholesteryl esters and monounsaturated fatty acids in the livers of the colonized mice.

From a clinical perspective, understanding the intricate relationship between systemic and microbial lipids may provide a unique avenue to study human metabolism. The fecal lipidome provides a non-invasive strategy to study the lipophilic activity of gut microbes, and their co-metabolism [14]. However, feces represent a highly heterogeneous sample matrix and, as such, pose several analytical challenges [15]. To extract useful information from stool samples, optimal sample preparation, handling, along with accurate and reliable analytical methods are required. Gregory et al. reported the lipidome composition of human stool samples for the first time [16]. The authors identified over 500 intact lipid species across six of eight different LIPID MAPS categories: glycerophospholipids, fatty acyls (FAs), SPs, glycerolipids, sterol lipids and prenol lipids [16]. Van Meulebroek et al. later optimized the lipidomics protocol for the analysis of stool samples to comprehensively map the lipidome [17]. More recently, Trošt et al. also screened fecal lipidome profiles from 10 healthy human volunteers [18]. The authors report that ceramides, DGs and TGs were the most abundant lipids found in stools. Lipids classes that are commonly detected at higher concentrations in plasma, e.g., lysophosphocholines, glycerophosphocholines and sphingomyelins were also present in the fecal samples, however, they were found at markedly lower concentrations [19].

One major challenge in fecal lipidome analysis is a largely technical one, i.e., the ability to identify and quantify the entire set of lipids in the stool. Processing of raw fecal lipidomics data shares common steps with mass spectrometric (MS)-based metabolomic analysis [20,21,22]. At present, identification of ‘unknown’ metabolites using tandem mass-spectrometry (MS/MS) is challenging, as limited number of reference spectra and/or authentic standards are available. Gao et al. [23] and Phua et al. [24] have identified metabolites from fecal samples using NIST MS libraries (www.chemdata.nist.gov). Several other commercial and non-commercial databases such as FiehnLib [25], Golm Metabolome Database [26], Human Metabolome Database (HMDB) [27], LIPID MAPS [28] and METLIN [29], have enabled identification of metabolites, including lipids. Additionally, in silico tools have also been developed to facilitate lipid identification, including structure database-dependent methods and spectra library-dependent methods. Spectra library independent tools such as LipidHunter, MS-DIAL and MZmine2 also provide specific workflows for profiling the lipidome. The integration of several metabolite databases might extend the coverage of lipids; however, this results in a markedly more complex workflow. CEU Mass Mediator [30] is a computationally-efficient integrative framework developed for this purpose; it can search for molecular lipids using multiple databases. However, to enable precise identification of lipids, both in silico as well as analytical advances are needed (reviewed in detail by Wei et al. 2019) [3].

Together, the fecal lipidome provides information regarding the metabolic interplay between the host, diet and gut microbiome [31]. An increasing number of studies suggest that the stool lipidome provides a functional readout of microbial metabolism; however, quantification of the lipids in the feces will likely result in a combination of host-originating, microbe-originating or host-microbiome co-metabolic products. Thus, whilst the fecal lipidome can suggests useful, plausible links between gut microbiota and host molecular lipids, the analysis of the fecal metabolome alone is incapable of distinguishing such origins. Bar et al. provides machine-learning algorithms to link factors such as human genetics, diet and microbiome with circulating small molecules in serum [6]. A similar approach may shed light on the factors that affect the origin of microbial small molecules in the gut, including the origin of microbial lipids. Here, the challenge would be to obtain large-scale measurements of several, potentially confounded variables (diet, microbiome, metabolome and other clinical variables), as well as the use of analytical methods for the capture of interactions (gene‑metabolite link) between variables [32]. Besides that, mechanistic studies are required to truly validate these associations [33]. In Section 3, we highlight several studies that mechanistically demonstrate gut microbiota-dependent lipid biotransformation, in particular showing the role of stool microbial species in the regulation of host cholesterol and sphingolipid homeostasis.

## 3. Synthesis of Lipids by Gut Microbes

Homeostasis between the molecular lipids and gut microbiota is vital for the host metabolic state. Despite a wide range of metabolite classes being produced by the gut microbiota [34,35], only a few metabolites have lipophilic characteristics, that is, they are able to pass through the epithelial barrier and thus directly impact the host metabolism. Gut microbiota composition and its derived lipids can impact the host metabolic state by altering plasma lipid levels. Here, we discuss specific classes of lipids of microbial or potentially-microbial origin, which are involved in the metabolic interplay between the gut microbiota and the host (Table 1).

### 3.1. Sphingolipids

SPs are bioactive lipids that regulate various cellular processes including cell differentiation, proliferation, apoptosis and inflammation [36,37]. In humans, some SPs can be obtained from the diet, while other species are generated by de novo synthesis. The biosynthesis of SPs has been extensively studied in eukaryotes [38]; however, recently the commensal gut microbes (*Bacteroides*, *Prevotella* and *Porphyromonas*) have been reported to produce SPs. Brown et al. reported these microbial derived SPs included ceramide phosphoinositol and deoxy-sphingolipids [33]. Intriguingly, these SPs are reported to aggravate intestinal inflammation and regulate host ceramide pools in animals. In addition, the authors found that microbially-derived SP deficiency was associated with inflammatory bowel disease (IBD), and impacted host-derived SP abundances in humans [33]. Similarly, Johnson et al. mapped the fate of dietary SPs in the gut microbiome, and showed that *Bacteroides*-derived SPs have an adverse effect on host SP metabolism, specifically on hepatic ceramide levels [39]. Intriguingly, using cell culture, they showed that microbially-derived SPs are incorporated into mammalian SP pathways. More recently, Lee et al. charted the pathways of dietary SPs in the gut microbiome (Figure 1) [40]. Taking the Bioorthogonal labeling-Sort-Seq-Spec (BOSSS) approach, the authors found that dietary SPs were mainly consumed by *Bacteroides.* They also found that *Bifidobacterium*, which do not produce SPs, could process dietary SPs in a manner similar to that of SP-producer *Bacteroides*, suggesting that bioactive lipids which are metabolically-accessible to the gut microbiome could be the target for the host to control its gut microbial composition [40].

### 3.2. Sterol

#### 3.2.1. Bile Acids and Derivatives

Primary BAs are produced in hepatocytes; however, secondary BAs such as deoxycholic acid (DCA) and lithocholic acids (LCA), ursodeoxycholate (UDCA) and numerous (≥50) others [41,42] are produced or biotransformed by gut microbiota. Briefly, in the distal ileum, the conjugated primary BAs are deconjugated by microbial bile salt hydrolases expressed predominantly by anaerobic intestinal bacteria of the genera *Bacteroides*, *Clostridium*, *Lactobacillus*, and *Bifidobacteria*. This is then followed by 7α-dehydroxylation by a bacterial 7α-dehydroxylase mainly expressed by *Clostridium* and *Eubacterium*. Further modifications include oxidation and epimerization of the hydroxyl groups by *Bacteroides*, *Clostridium*, *Escherichia*, *Eggerthella*, *Eubacterium* and *Peptostreptococcus* [43]. Under normal physiological conditions, secondary BA synthesis represents less than 10% of total BA synthesis. Similarly to primary BAs, secondary BAs also modulate host metabolism, the innate immune system in addition to acting as signaling molecules [35,44]. Ridlon et al. extensively reviewed the emerging BA-gut-microbiome axis [45]. A summary of the key BAs and their microbial linkage are presented in Table 1. 

#### 3.2.2. Cholesterol

Cholesterol is a key precursor molecule in the synthesis of many different lipids, including BAs and Vitamin D (fat-soluble secosteroids) [63]. Circulating cholesterol is either derived from the diet or produced by de novo synthesis in hepatocytes. Since cholesterol from both sources pass through the intestine, it has been suggested that the gut microbiota modulates plasma cholesterol levels [64,65]. Recently, by integrating metabolomics and shotgun metagenomics data, Kenny et al. identified the gut microbial enzymes involved in cholesterol metabolism (Figure 2) [52]. They found that the cholesterol dehydrogenases enzyme (encoded by *ismA* genes of the gut microbiota) has a significant impact on both fecal and total serum cholesterol levels in humans [52]. However, whether these metabolically-active gut microbes are linked to the diet or not remains unknown. In the Dutch LifeLines-DEEP cohort involving 893 human subjects, Fu et al. showed that the gut microbiome contributed a substantial proportion of the variation in circulating high-density lipoprotein cholesterol (HDL), but not to total cholesterol or low-density lipoprotein (LDL) cholesterol levels [9]. Taken together, these studies suggest that cholesterol metabolism by gut microbes is crucial in host cholesterol homeostasis (Figure 2) [52]. 

### 3.3. Fatty Acyls and Conjugates

Fatty acids produced by gut microbiota can stimulate synthesis of mono-unsaturated fatty acids (MUFAs) and the elongation of polyunsaturated fatty acids (PUFAs) in the host. A multi-omics profiling study found that circulating lipid levels can be affected by microbial fatty acid metabolism [66]. MUFA (16:1*n*−7, FA 18:1*n*−9, 18:1*n*−7), MUPC (PC 34:1, PC 36:1) and PUFA (20:3*n*−6, 22:6−3) levels were associated with acetate production in the gut. In addition to their free forms, fatty acids can also be found conjugated to mono-amine neuro-transmitters, such as serotonin, forming arachidonoyl-serotonin (AA-5-HT), oleoyl-serotonin, palmitoyl-serotonin, and stearoyl-serotonin. The neurotransmitter serotonin (5-hydroxytryptamine) in humans is mainly (90%) synthesized in the gastrointestinal tract by human gut microbiota [67] (see also related Section 3.4 below, on endocannabinoids). Serotonin and SCFA concentrations stimulate the formation of *N*-acyl ethanolamine (NAE) conjugates [68], which form a novel class of endogenous lipid mediators in the intestine. Emerging evidence also indicates a role for the intestinal microbiota in the production of *N*-acyl serotonin’s metabolites, which have been shown to modulate the enteric nervous system [69,70]. Additionally, conjugated fatty acids are also formed by commensal gut microbes such as *Lactobacillus plantarum* [53]. In particular, conjugated linoleic acids (CLAs), such as conjugated diene structures *cis*-9, *trans*-11-CLA and *trans*-9,*trans*-11-CLA, are formed in the gastrointestinal tract passing through the intermediate 10-hydroxy-12-octadecenoic acid [53].

### 3.4. Endocannabinoids

The endocannabinoid system is comprised of lipid-derived endogenous cannabinoid receptor ligands (endocannabinoids), enzymes involved in their synthesis and degradation, and the cannabinoid 1 and 2 receptors (CB1R and CB2R), which have differential affinities for endocannabinoids. Growing evidence suggests that gut microbes and the endocannabinoid system are interlinked [58,71]. Rousseaux et al. initially proposed the link between the gut microbes (*Lactobacillus acidophilus*) and the endocannabinoid system [71]. Later, Cani et al. demonstrated indirect crosstalk between the gut microbiota and the endocannabinoid system, which modulated host adipogenesis [72]. The authors found that peripheral (e.g., intestine and adipose tissue) endocannabinoid (specifically, anandamide; AEA) levels were influenced by gut microbiota in the experimental animals. In another study, Everard et al. showed that *A. muciniphila* (which represents 3–5% of the human microbial community) treatment in obese mice increased the levels of glycerolipids intestinal 2- oleoylglycerol (2-OG), 2-arachidonoylglycerol (2-AG) and 2- palmitoyl-glycerol (2-PG) [73]. More recently, endocannabinoid–like molecule *N*-acyl-3-hydroxypalmitoyl-glycine (commendamide) was reported to be produced by the commensal microbe *Bacteroides* [74,75]. These findings suggest that specific gut microbes produce specific classes of lipids which impact these host signaling and metabolic pathways.

### 3.5. Carnitine and Acyl Carnitines

De novo synthesis of carnitines occurs in all domains of life [76]. Carnitines that are not digested or absorbed in the small intestine are excreted to the large intestine, where they are catabolized by gut microbes. Some microbes (*Pseudomonas* species) utilize carnitine as a source of carbon and nitrogen, while *Acinetobacter calcoaceticus* catabolize carnitine into trimethylamine (TMA), which is then converted into trimethylamine N-oxide (TMAO) by hepatic flavin monooxygenases (FMOs). TMAO is strongly linked with cardiovascular diseases [77,78]. Also, phosphatidylcholines (PCs), a major a class of phospholipids, forming the major structural components of cellular plasma membranes are also converted to trimethylamine (TMA) by gut microbes. These PCs are abundant in foods such as fish, eggs and milk, suggesting that dietary lipids regulate host lipid metabolism through interaction with the gut microbiota [79]. Interestingly, Hulme et al. identified two structural analogs of carnitine, 3-methyl-4-(trimethylammonio) butanoate and 4-(trimethylammonio) pentanoate, which are produced by anaerobic commensals from the *Clostridiales* family [80]. Acyl carnitines (fatty acyl esters of L-carnitine) have also been reported in human feces [81,82]. However, information about the microbes that facilitate the biotransformation of acyl carnitines in the gut is still unknown [52,58,80].

## 4. Functional Profiling and Metabolic Modeling of Human Gut Microbiome for Understanding Microbial-Host-Lipid Co-Metabolism

At present, there are many tools, strategies and approaches available for the integration of microbiome–metabolome data [83]. In recent years, whole genome shotgun metagenomic sequencing (WGS) [84] has been used for phylogenetic and functional profiling of the gut microbiome [85]. WGS datasets have aided in the identification of microbial dysbiosis in metabolic disorders [8,86] and helped to observe strain-level perturbations of the gut microbiota by fecal microbiome transplantation (FMT) [87]. Several bioinformatics pipelines, e.g., MOCAT2 [88], Metagenomics Rast (MG-RAST) [89], HMP Unified Metabolic Analysis Network (HUMAnN2) [90], were designed for microbiome profiling and functional analysis. In addition, MEtaGenome ANalyzer (MEGAN) [91], CAMERA [92] and GALAXY (https://usegalaxy.org/, accessed on 10 December 2020) provide standalone and/or web-based platforms to analyze large-scale metagenomics datasets. A detailed overview of metagenomics data acquisition and processing is reviewed in [93]. Recently, deep learning (DL)-based neural networks, as applied to WGS data, have improved gene prediction [94], and phylogenetic classification [95] of human gut microbiota. DeepMicrobes, a DL-based framework was trained on the bacterial repertoire of the human gut microbiome [96]. The DL model outperformed (fewer false positives) the state-of-the-art taxonomic classification tools for microbiome profiling (identification and quantification), at species and genus levels. In addition, DeepMicrobes have identified several novel microbial signatures in inflammatory bowel disease [95].

On the other hand, amplicon-based metataxonomic sequencing has been used to stratify microbes at taxonomic levels (Operational Taxonomic Unit, OTU) [97]. OTUs represent the phylogenetic diversity and microbial richness, between or among the samples and/or the environment [98,99]. Several curated taxonomy databases such as Greengenes (16S) [100], Silva (16S + 18S) [101] and Unite (ITS) [102] have been developed, and these provide full-length 16S rRNA reference gene sequences. Computational tools such as QIIME2 [103] and MEGAN [91] are commonly used for 16S rRNA analysis. Metataxonomic data analysis is reviewed elsewhere [104].

Together, WGS and amplicon sequencing [105] can be used for microbiome profiling. In addition, WGS data can be used for functional characterization of gut microbiota. Furthermore, WGS can be used for gene discovery [106]. In contrast, amplicon-based metataxonomic sequencing is biased by the high copy number of marker genes [107]. Moreover, it lacks the ability to characterize the genes’ functions (e.g., metabolic, immune) in the gut community [107,108]. Neither of these techniques can provide a mechanistic overview of gut microbial metabolism and their underlying processes. To study the intricate relationships between microbial genes and molecular lipids, in the context of metabolic pathways, taxon-specific metabolic reconstructions of microbes (provided the genome is sequenced) have been developed. These genome-scale microbial reconstructions and models have enhanced our understanding of host-microbiome interactions (e.g., co-regulation) in healthy vs. disease conditions [109,110].

Therefore, genome-centric methods might not provide mechanistic insight into the interactions of microbial species or strains in a gut ecosystem. To understand the intricate relationship between gut microbes, diet and their metabolic interplay with the host, several computational models have been developed [83,109,110,111,112]. Among various computational modeling approaches, genome-scale metabolic modeling (GSMM), which is a constraint-based mathematical modeling approach, has enhanced our understanding of host-microbiome interactions under different clinical conditions. Moreover, genome-scale metabolic models (GEMs) of gut microbes have provided testable hypotheses on diet-microbe-host axis interactions in healthy vs. disease states [109,113,114,115]. Several tools such as Kbase [116], ModelSEED [117], COnstraint-Based Reconstruction and Analysis (COBRA) [118] and RAVEN (Reconstruction, Analysis, and Visualization of Metabolic Networks) [119], have enabled and/or aided in the reconstruction of microbial-GEMs from available genomic and metagenomic data. A detailed overview of GSMM, as applied to human gut microbiota, is reviewed elsewhere [109,110].

As an example, GSMM was used to study metabolic interactions in the diet-microbiota-host axis in 45 obese and overweight individuals [114]. GSMM was able to estimate the metabolic capabilities and fluxes of the gut microbes for these groups of individuals. In addition, it predicted an abrupt change in amino acid and short-chain fatty acid (SCFA) fluxes in response to a dietary intervention. The results from GSMM were validated by fecal and blood metabolomics data. Furthermore, GSMM has also been used to elucidate the metabolic pathway(s) of human gut microbes in malnourished children from Bangladesh and Malawi. Here, GSMM identified a significant increase in intestinal butyrate production which was associated with a pair of gut microbes [120]. In another study, GSMM was used to study BA biotransformation by human gut microbiota [115]. The results here showed that *Bacteroides* and *R. gnavus* species can, together, produce ursodeoxycholate (UDCA) [115]. This work was performed using AGORA (Assembly of Gut Organisms through Reconstruction and Analysis); AGORA is a compendium of semi-curated metabolic reconstructions of the human gut microbiota [121]. Microbial GEMs are publicly available via Virtual Metabolic Human (VMH) [122] and the BiGG [123] database.

However, modeling the lipid metabolism in human gut microbiota on a genome-scale poses several challenges. Firstly, mapping of the experimentally-measured lipidome data on to the genome-scale metabolic network is highly challenging, and as such, it limits to constrain an individual lipid reaction of a GEM towards production of biomass, and thus, making the model predictions poorly comparable with the experimental data. Different identifiers or annotations of the molecular lipids are also used in different experimental datasets and in GEMs. GEMs assemble individual lipids (e.g., PC (45:0), and SM (36:1)) into a pool of their generic classes (e.g., PCs and SMs). Therefore, it is necessary to annotate the lipids both in the experimental data and in the GEMs based on their ontology, and chemical identifiers *(InChl, SMILES)*. Recently, Poupin et al. suggested a matching strategy using ChEBI ontology to bridge the gap between lipidomics data and genome-scale metabolic network (GSMN). Ontology-based mapping provided intricate links between generic classes of lipids present in the GSMN, with its congener molecular species identified in the lipidomics datasets [124].

Recently, several tools such as LION/web [125], LipidLynxX [126] and Reference Set of Metabolite Names (RefMet) [127] have been developed with a view to solving these issues. LION/web [125] aims to translate and interpret the involvement of lipid species in various biological systems. LipidLinxX is a data transfer hub which facilitates integration of large-scale lipidomic datasets by standardizing lipid identifiers to the same level of annotation and thereby facilitate cross-level matching between different datasets [126]. RefMet seeks to provide unifying nomenclature for the metabolites detected by analytical methods [127]. Furthermore, Gerhard Liebisch et.al. [128] proposed a comprehensive and standardized system to report lipid structures analyzed by MS data. In the future, such guidelines should be followed for the reporting of lipidomics data which, in turn, can improve the integration of lipids into GSMNs.

Additionally, the Split Lipids Into Measurable Entities reactions *(SLIMEr)* [128] tool has the ability to divide lipid species into their respective classes and compute their acyl chain carbon distributions. Therefore, it can constrain a lipid pathway when given the lipid classes and acyl chain distributions of its substrates. *SLIMEr* was developed to determine the biomass requirements of lipids in *S. cerevisiae*. Although these approaches have improved integration of lipidomics data into GSMNs [124,128], the mapping of lipidomics data still requires extensive harmonization between individual lipid species annotated in the GSMN and in the lipidomics datasets [124]. Even so, some lipid pathways in the gut microbes remain uncharacterized due to a lack of bibliographic evidence and/or experimental data.

## 5. Conclusions and Future Perspectives

Human gut microbiota metabolize, transform and hydrolyze complex lipids, which, in turn, modulate host lipid homeostasis, and thereby affect other physiological processes. Fecal lipidomic analysis may thusly reveal causal slink between gut microbes and circulating lipids. However, identification of potential microbial lipids in the stool lipidome remains a challenging task, and is limited by both analytical techniques (due to complexity of stool specimen) and annotation of lipids from the metabolite databases (due to lack of microbial-lipid specific databases). Recently, several studies using animal models, as well as in humans, have identified an interplay between microbial lipids, systemic lipids and states of disease. It is also evident that different gut microbial species are involved in lipid metabolism. Furthermore, it remains to be investigated as to how combinations of lipids and dietary fiber affects gut microbial composition and, subsequently, host systemic lipid levels., Gut microbial lipid pathways are not fully characterized, owing to the lack of metagenomic annotation or reference genomes for ‘uncultured’ or ‘unknown’ microbes in the human gut. We envisage that advancements in the field of GSMM, together with emerging mechanistic studies and meta-‘omics’ techniques will aid the study of host-microbial lipid co-metabolism across different health and disease states.

## Figures and Tables

**Figure 1 metabolites-11-00055-f001:**
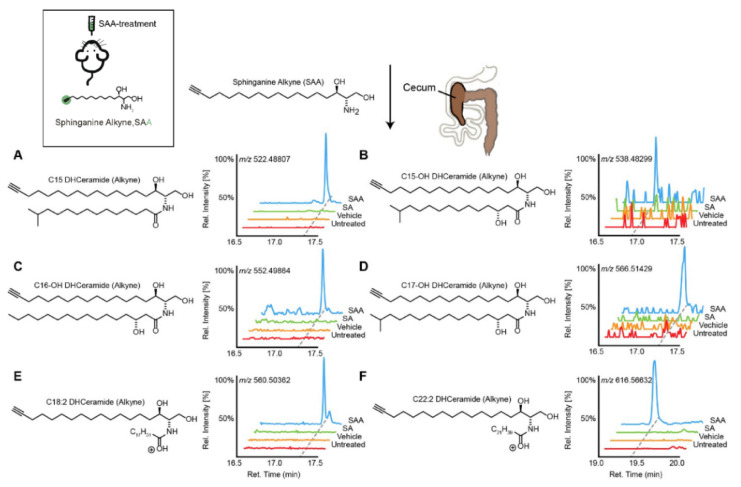
Transformation of dietary sphinganine into dihydrocermides by the gut microbes. Here, sphinganine alkyne (SAA) was given to mice by oral gavage (five consecutive days), then the fecal content from mice was collected and metabolic consequences of SAA exposure were determined using high-resolution mass spectrometry ion chromatograms. The authors showed a distinct cecal lipidome chromatograms for mice that orally treated with SAA (blue), which contained the alkyne-bearing (**A**) C15-, (**B**) C15OH-, (**C**) C16OH-, (**D**) C17OH-, (**E**) C18:2- and (**F**) C22:2- dihydrocermides. However, these dihydrocermides were absent in treatments with sphinganine (SA, green), vehicle or no treatment (red). Figure adapted from [40], with permission under CC BY 4.0 license.

**Figure 2 metabolites-11-00055-f002:**
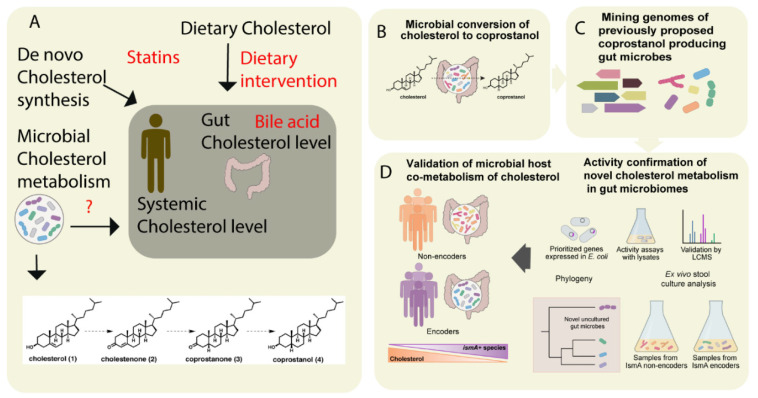
The metabolism of cholesterol by gut microbes influences both intestinal and circulating cholesterol concentrations. (**A**) The cholesterol level in the blood can be influenced either by de novo cholesterol synthesized in the liver or it may be derived from exogenous sources such as the diet. The endogenous cholesterol level may be also influenced by drugs such as statins, or via altered bile acid metabolism. In addition, gut microbial metabolism of cholesterol may also serve as check point for the maintenance of cholesterol homeostasis (**B**). As shown in panel **A**, the authors proposed a pathway for microbial conversion of cholesterol (1) to coprostanol (4) in the microbiota involves the intermediates cholestenone (2) and coprostanone (3). (**C**) By using genome mining, Kenny et al. identified and characterized microbial enzymes linked with cholesterol metabolism. Later, integrating the functional metagenomics and metabolomics data, they predicted as well as validated a group of microbes and their microbial activities (microbial cholesterol dehydrogenases) which mediated the metabolism of cholesterol in the gut (**D**). Figure adapted from [52] with permission under CC BY 4.0 license.

**Table 1 metabolites-11-00055-t001:** Significant associations of bioactive lipids, gut microbiome related classes and lipid classes.

Lipid Category	Lipid Sub Class	Example/Related Lipids	Microbes	References
**Sphingolipids**	Ceramide phosphoinositols	PI-Cer(d18:1/22:0)	*Bacteroidetes* (*genera Bacteroides*, *Prevotella*, *Porphyromonas*, *Bacteroides theta*, *thetaiotaomicron*, *ovatus and fragilis*	[33]
	Ceramide phosphoethanolamines	N-Acyl ceramide phosphoethanolamine		
	Sphinganines	3-ketosphinganine sphinganine		
	*N*-acylsphinganines	dihydroceramide		
		C15-, C15OH-, C16OH-, C17OH-, C18:2-, C22:2- dihydrocermide		[40]
	Sphingoid base 1-phosphates	sphinganine-1-phosphate (d17:0) sphinganine-1-phosphate (d18:0)		[39]
**Sterol**	C24 bile acids, alcohols, and derivatives	deoxycholic acid lithocholic acid ursodeoxycholate iso-deoxycholic acid iso-lithocholic acid 7-oxo-lithocholic Acid	*Bacteroides*, *Clostridium*, *Lactobacillus*, and *Bifidobacteria* and *Alloscardovia* sp.	[15,46,47,48,49] [41,50]
	Taurine conjugates	tauroursodeoxycholic acid		
	Cholesterol and derivatives	cholestenone coprostanone coprostanol	*Eubacterium coprostanoligenes*, *Bacteroides intestinalis*, *Faecalibacterium prausnitzii*	[51,52]
**Fatty Acyls**	Other Octadecanoids	10-hydroxy-12 (Z)-octadecenoic acid (18:1) (HYA), 10-hydroxy-12,15(Z,Z) octadecenoic acid (18:2) (αHYA), 10-hydroxy-6,12(Z,Z)-octadecadienoic acid (18:2) (γHYA), 10-hydroxyoctadecanoic acid (HYB), 10-hydoroxy-trans-11-octade-cenoic acid (HYC), 10-oxo-12(Z)-octadecenoic acid (18:1) (KetoA), 10-oxo-12,15(Z,Z) (18:2) octadecenoic acid (αKetoA), 10-oxo-6,12(Z,Z)-octadecenoic acid (18:2) (γKetoA), 10-oxo-octadecanoic acid, 10-oxo-trans-11-octadecenoic acid	*Lactobacillus genus* (*Lactobacillus plantarum*, *Lactobacillus salivarius*, *Lactobacillus gasseri*, *Lactobacillus acidophilus* and *Lactobacillus johnsonii*) *Bifidobacterium* spp., *Eubacterium ventriosum* and *Lactobacillus* spp.	[53,54,55,56,57]
	Unsaturated fatty acids	oleic acid		
**Glycerolipids (*Endocannabinoid*)**	Monoacylglycerols	2-arachidonoylglycerol (2-AG) 2- oleoylglycerol (2-OG) 2- palmitoyl-glycerol (2-PG)	Akkermansia muciniphila	[58,59,60,61,62]

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
