# Peer review of "Linking Gut Microbiome and Lipid Metabolism: Moving beyond Associations"

_metabolites, 2021, doi:10.3390/metabo11010055_

Round 1

Reviewer 1 Report

In the manuscript by Santosh Lamichhane et.al. entitled "Linking Gut Microbiome and Lipid Metabolism: Moving Beyond Associations“, the authors reivewed major points from identification of lipidome/metabolome till the GSMN integration of the data and sumarized the literatures of lipid of microbial or potentially micobial origin by major lipid classes. The manuscript also provide a important section suggesting the modeling lipid metabolism in human gut microbiota on a genome-scale and addressed few main issues.

The manuscript is well structured and I suggested to accept after minor revision and following comments need to be corrected prior to publication.

Comments:

According to LIPID MAPS definition, the abbreviation of sphingolipids should be SP and SL is assigned to Saccharolipids (https://www.lipidmaps.org/data/classification/LM_classification_exp.php). The abbreviation SP should be used in the revised version to represent sphingolipids through the whole manuscript.

Line 93-104 provided a summary for identification of metabolites including lipids using spectra libraries. However, recent years there are several lipid specific identification tools using different approaches such as database independent methods, structure database dependent methods, and spectra library dependent methods. Tools such as LDA, LipidHunter, MS-DIAL, MZmine2 and several others can be mentioned. It is important to address in this review that there are lipid specific workflows, especially the spectra library independent tools for the profiling of lipidome. The combination of general metabolomics workflows and lipid specific tools is essential to obtain the overall metabolite and lipid profile which may further be used for GSMM.

Table 1

Cholesterol row: There is a reference in PMID which should be converted into citation format.

Sphingolipids row: There is a “phosphoethanolamides” in the second column, which I suppose it should be “phosphoethanolamine” and according to LIPID MAPS definition. An example lipid from LIPID MAPS: LMSP03020001 (Category: Sphingolipids [SP], Main Class: Phosphosphingolipids [SP03], Sub Class: Ceramide phosphoethanolamines [SP0302]). However, the “Phosphosphingolipids” in this table has a separate row below.

Glycerophosphoethanolamines row: “Phosphorylethanolamine” is a single compound (C2H8NO4P) while the reference 32 mentioned multiple lipids in the Glycerophosphoethanolamines (PE) sub class. “Phosphorylethanolamine” should be replaced by “Glycerophosphoethanolamines” or “Phosphatidylethanolamine” which represent the lipids in this sub class.

The Lipid column in this table is using mixed lipid sub classes and lipid individual lipids and the lipid class column contains mixed levels of lipid categories (Sphingolipids), class(Glycerophosphoethanolamines) and sub classes (Phosphonosphingolipids).

The sections 3.1 to 3.6 reviewed lipids of micobial or potentially microbial origin. However Glycerophosphoethanolamine sub class, which in the table 1, is missing in the text.

I suggest to reformat this table carefully according to the definitions provided by LIPID MAPS and change the headings from “lipid” to something like “example/related lipids”. The authors have to decide on which information level they what to summarize the information. In general, a structure such as lipid category/main class → lipid sub class → example/related lipids should be followed. I would be better if the table rows fit to the sections and order of sub sections in section 3, so that readers can follow/find the reference easier. e.g. row Fatty acids and conjugates and row Fatty Acyls/ PUFA-derived can be merged to fit section 3.4.

I found papers such as PMID: 23518648 described “It is likely that Bacteroides enzymes hydrolyse dietary PC(Glycerophosphocholines), contributing to the total choline/TMA load in the host.” I think it would be better to add few paragraphs to give a brief review of how the host lipid/diet lipid which undertaken by gut microbiota and further involved in the interplay between the gut microbiotaand the host since it is also mentioned in line 272-273.

Type errors such as line 302 “...Therefore, it is are necessary…” should be checked and corrected.

Line 304: SMILES is a text notation of the chemical structure without fix lenth similar to InChI (but not InChiKeys) which can be converted to the structure of the molecule rather than a fixed length code that can be used as identifier in database. I think the authors mean the exact chemical structure notation here, so it would be “chemical structure notations (InChI, SMILES)”

Line 301-316 describe several lipid class and lipid species mapping issues, this year there are several tools published aiming to solve some of issues such as LION/web, LipidLynxX and RefMet which can be mentioned. The Lipid nomenclature issues is a related topic which can be addressed briefly. Recent effort to standardize the lipid shorthand notation has been published by Gerhard Liebisch et.al. PMID: 33037133, such guidelines should be followed in future report of lipidomics data to improve the integration of lipids into GSMN.

It would nice if authors can write few sentences about if it is possible/necessary or not to connect the GSMN of multiple gut microbiota species and GSMN of the host to simulate the flux of metabolites and lipids in the grand network and contribute to the understanding of the interplay between gut microbiota and host.

There is no section 5 in the PDF I downloaded. I suppose the section 6 is miss labeled and should be section 5.

Author Response

In the manuscript by Santosh Lamichhane et.al. entitled "Linking Gut Microbiome and Lipid Metabolism: Moving Beyond Associations“, the authors reivewed major points from identification of lipidome/metabolome till the GSMN integration of the data and sumarized the literatures of lipid of microbial or potentially micobial origin by major lipid classes. The manuscript also provide a important section suggesting the modeling lipid metabolism in human gut microbiota on a genome-scale and addressed few main issues.

The manuscript is well structured and I suggested to accept after minor revision and following comments need to be corrected prior to publication.

Comments:

According to LIPID MAPS definition, the abbreviation of sphingolipids should be SP and SL is assigned to Saccharolipids (https://www.lipidmaps.org/data/classification/LM_classification_exp.php). The abbreviation SP should be used in the revised version to represent sphingolipids through the whole manuscript.

RESPONSE: We agree with the reviewer. The abbreviation SL is replaced with SP in the revised manuscript.

Line 93-104 provided a summary for identification of metabolites including lipids using spectra libraries. However, recent years there are several lipid specific identification tools using different approaches such as database independent methods, structure database dependent methods, and spectra library dependent methods. Tools such as LDA, LipidHunter, MS-DIAL, MZmine2 and several others can be mentioned. It is important to address in this review that there are lipid specific workflows, especially the spectra library independent tools for the profiling of lipidome. The combination of general metabolomics workflows and lipid specific tools is essential to obtain the overall metabolite and lipid profile which may further be used for GSMM.

RESPONSE: Thank you very much for pointing this out. The Line 93-104 has been modified as per context suggested by the reviewer.

Cholesterol row: There is a reference in PMID which should be converted into citation format.

RESPONSE: This has been updated in the revised manuscript.

Sphingolipids row: There is a “phosphoethanolamides” in the second column, which I suppose it should be “phosphoethanolamine” and according to LIPID MAPS definition. An example lipid from LIPID MAPS: LMSP03020001 (Category: Sphingolipids [SP], Main Class: Phosphosphingolipids [SP03], Sub Class: Ceramide phosphoethanolamines [SP0302]). However, the “Phosphosphingolipids” in this table has a separate row below.

RESPONSE: Thank you very much for pointing this out. This has been updated in the revised manuscript.

Glycerophosphoethanolamines row: “Phosphorylethanolamine” is a single compound (C2H8NO4P) while the reference 32 mentioned multiple lipids in the Glycerophosphoethanolamines (PE) sub class. “Phosphorylethanolamine” should be replaced by “Glycerophosphoethanolamines” or “Phosphatidylethanolamine” which represent the lipids in this sub class.

RESPONSE: Thank you very much for pointing this out. The Table has been updated in the revised manuscript.

The Lipid column in this table is using mixed lipid sub classes and lipid individual lipids and the lipid class column contains mixed levels of lipid categories (Sphingolipids), class (Glycerophosphoethanolamines) and sub classes (Phosphonosphingolipids).

RESPONSE: Thank you very much for pointing this out. The Table has been updated in the revised manuscript.

The sections 3.1 to 3.6 reviewed lipids of micobial or potentially microbial origin. However Glycerophosphoethanolamine sub class, which in the table 1, is missing in the text.

RESPONSE: Thank you very much for pointing this out. The Table has been updated in the revised manuscript.

I suggest to reformat this table carefully according to the definitions provided by LIPID MAPS and change the headings from “lipid” to something like “example/related lipids”. The authors have to decide on which information level they what to summarize the information. In general, a structure such as lipid category/main class → lipid sub class → example/related lipids should be followed. I would be better if the table rows fit to the sections and order of sub sections in section 3, so that readers can follow/find the reference easier. e.g. row Fatty acids and conjugates and row Fatty Acyls/ PUFA-derived can be merged to fit section 3.4.

RESPONSE: Thank you very much for pointing this out. The Table has been updated in the revised manuscript.

I found papers such as PMID: 23518648 described “It is likely that Bacteroides enzymes hydrolyse dietary PC(Glycerophosphocholines), contributing to the total choline/TMA load in the host.” I think it would be better to add few paragraphs to give a brief review of how the host lipid/diet lipid which undertaken by gut microbiota and further involved in the interplay between the gut microbiota and the host since it is also mentioned in line 272-273.

RESPONSE: This is a relevant comment by the reviewer, we have added few sentence in section 3.6 as well as in section 4. In addition, the relationship between the gut microbiome, host lipids / metabolites and genetics are discussed in line [111-126]. As a case, we stated the interplay between diet–microbiome in lean and obese individuals using GSMM, which is stated [lines: 306-318]. A relationship between host lipids / BA biotransformation facilitated by gut microbes is discussed in lines [314-315].

Type errors such as line 302 “...Therefore, it is are necessary…” should be checked and corrected.

RESPONSE: Many thanks for pointing this out, this typo has been corrected.

Line 304: SMILES is a text notation of the chemical structure without fix lenth similar to InChI (but not InChiKeys) which can be converted to the structure of the molecule rather than a fixed length code that can be used as identifier in database. I think the authors mean the exact chemical structure notation here, so it would be “chemical structure notations (InChI, SMILES)”

RESPONSE: This information has been corrected and updated in the revised manuscript.

Line 301-316 describe several lipid class and lipid species mapping issues, this year there are several tools published aiming to solve some of issues such as LION/web, LipidLynxX and RefMet which can be mentioned. The Lipid nomenclature issues is a related topic which can be addressed briefly. Recent effort to standardize the lipid shorthand notation has been published by Gerhard Liebisch et.al. PMID: 33037133, such guidelines should be followed in future report of lipidomics data to improve the integration of lipids into GSMN.

RESPONSE: We thank the reviewer for the suggestion. We extended the paragraph and included relevant references as suggested [line: 331-339].

It would nice if authors can write few sentences about if it is possible/necessary or not to connect the GSMN of multiple gut microbiota species and GSMN of the host to simulate the flux of metabolites and lipids in the grand network and contribute to the understanding of the interplay between

RESPONSE: Recently, we published a review article (PMID: 30695998) which discuss about the community modelling of human gut microbiota using a GSMM approach. There we discussed about integrating various microbial models, and host–microbiome modelling. To avoid redundant information, we stated in [line 301-303], A detail overview of GSMM applied to human gut microbiota is reviewed elsewhere..” and also cited other articles in line. We provided an example of integrative microbial–GSMM that identified SCFAs and bile acids, which was validated by blood and stool metabolomics data [line 304-312].

There is no section 5 in the PDF I downloaded. I suppose the section 6 is miss labeled and should be section 5.

RESPONSE: Yes, that is correct. The section has been updated.

Reviewer 2 Report

Remarks to the Author:

In this manuscript, Lamichhane et al.  reviewed the state of the art in the field of gut microbiome and lipid metabolism. The authors present the available computational tools and databases for the detection and structural annotation of microbial lipids as well as the dissection tools elucidating their relationships to microbial communities. The review manuscript is well-written and informative; however, a few concerns need to be addressed before further consideration for publication.

Comments:

(1) The summarized approach presented in this manuscript is incapable of distinguishing whether the identified lipids are species-specific (solely host-origin, microbe-origin, or combined origin). The manuscript will be more educational if the authors could offer their suggestive perspectives for which concepts or relevant methodologies to apply to decipher precise origins of detected lipids.

(2) The current version of manuscript lacks reviewing references on approaches for functional annotation of the discovered microbial lipids facilitating their contextualization in physiological or biological pathways and networks.

(3) In general, the authors are encouraged to discuss strengths and limitations of the various technical approaches to semi-quantify/quantify lipids as well as the bioinformatics tools for analyzing the lipidome in relation to both microbiome data and host bio-clinical variables.

(4) In the manuscript, though the authors have stated the major challenge in faecal lipidome from a technical perspective, currently there are available tools that can be applied for a robust prediction of lipidome (PMID: 33177712). Considering adding sentences describing the lipidome predictive tools based on gut microbiome data is suggested.

(5) Authors need to appreciate and comment on confounders in modelling microbiome-lipidome data, as the regulatory check points in their accumulation/ processing/ diversity are difficult to model, i.e., age or sex or medications affecting lipidome or lipidome can be very different- all sorts of directionality and unknown interactions. How does one model those trends? 

(6) Moderate linguistic refinements/ corrections are desirable. Present version has some common, punctuation, tense, grammatical, typological sentence framing/ phrase construction issues which should be taken care by the authors, and some loosely constructed sentences floating around.

Author Response

In this manuscript, Lamichhane et al.  reviewed the state of the art in the field of gut microbiome and lipid metabolism. The authors present the available computational tools and databases for the detection and structural annotation of microbial lipids as well as the dissection tools elucidating their relationships to microbial communities. The review manuscript is well-written and informative; however, a few concerns need to be addressed before further consideration for publication.

Comments:

(1) The summarized approach presented in this manuscript is incapable of distinguishing whether the identified lipids are species-specific (solely host-origin, microbe-origin, or combined origin). The manuscript will be more educational if the authors could offer their suggestive perspectives for which concepts or relevant methodologies to apply to decipher precise origins of detected lipids.

RESPONSE: This is a relevant comment by the reviewer. We have update the manuscript addressing the concern from the reviewer (particularly section 2 of revised manuscript Line 110 - 125).

(2) The current version of manuscript lacks reviewing references on approaches for functional annotation of the discovered microbial lipids facilitating their contextualization in physiological or biological pathways and networks.

RESPONSE: This is a relevant comment by the reviewer, however we have discussed this context in section 3 of the current manuscript. The study by Johnson et al., Lee et al. 2020, as well  Brown et al focused on discovered microbial lipids and involved microbial species in particularly in the regulation of host cholesterol and sphingolipid homeostasis . To our best understanding these study provides physiological and biological pathways level information (lipid specific). Further, section 4 provides explanation and perspective to connect these pathway via network (i.e. genome scale).

(3) In general, the authors are encouraged to discuss strengths and limitations of the various technical approaches to semi-quantify/quantify lipids as well as the bioinformatics tools for analyzing the lipidome in relation to both microbiome data and host bio-clinical variables.

RESPONSE: We extended the discussion of lipidomics data processing [line 111-126] and provided a case study, where bioinformatics / machine learning method was used for the integrative analysis of lipidome­–microbiome in relation to host genetics. In this context, we also addressed various challenges and technicalities.

(4) In the manuscript, though the authors have stated the major challenge in faecal lipidome from a technical perspective, currently there are available tools that can be applied for a robust prediction of lipidome (PMID: 33177712). Considering adding sentences describing the lipidome predictive tools based on gut microbiome data is suggested.

RESPONSE: This is a relevant suggestion by the reviewer. We have updated out manuscript as per the context suggested by the reviewer (Line 112-116).

(5) Authors need to appreciate and comment on confounders in modelling microbiome-lipidome data, as the regulatory check points in their accumulation/ processing/ diversity are difficult to model, i.e., age or sex or medications affecting lipidome or lipidome can be very different- all sorts of directionality and unknown interactions. How does one model those trends? 

RESPONSE: This is now added in [line 111-126].

(6) Moderate linguistic refinements/ corrections are desirable. Present version has some common, punctuation, tense, grammatical, typological sentence framing/ phrase construction issues which should be taken care by the authors, and some loosely constructed sentences floating around.

RESPONSE: Thank you very much for this comment. This manuscript has been edited by Dr. Aidan McGlinchey who is a native English speaker. We have acknowledged Dr. Aidan McGlinchey.

Reviewer 3 Report

This provides a detailed review of the status of current research to understand the relationship between the microbiome and host lipids. The review focuses on the influence of the gut microbiome on the human lipidome including reference to relevant animal studies which support likely microbial metabolic pathways described and specific lipids implicated in effecting human health. The authors also include a description of strategies and challenges to determine human fecal lipidome. Additionally a comprehensive overview of metabolome and microbiome analysis and discussion of the strengths and weaknesses of approaches and applications of profiling and modelling techniques is included. Thus providing a good broad introduction for a non-expert reader. Although this review focusses on proposed roles of microbiome in impacting on human host lipid constituents and the potential link to disease, it is acknowledged that diet and host genetics will have further influence which is a crucial point for the context of the research presented.

Minor typos identified: line 156 suggest ‘Similar to’; line 185 ‘ suggest ‘validated’

Author Response

This provides a detailed review of the status of current research to understand the relationship between the microbiome and host lipids. The review focuses on the influence of the gut microbiome on the human lipidome including reference to relevant animal studies which support likely microbial metabolic pathways described and specific lipids implicated in effecting human health. The authors also include a description of strategies and challenges to determine human fecal lipidome. Additionally a comprehensive overview of metabolome and microbiome analysis and discussion of the strengths and weaknesses of approaches and applications of profiling and modelling techniques is included. Thus providing a good broad introduction for a non-expert reader. Although this review focusses on proposed roles of microbiome in impacting on human host lipid constituents and the potential link to disease, it is acknowledged that diet and host genetics will have further influence which is a crucial point for the context of the research presented.

RESPONSE: The context of diet host and microbe interaction is a relevant comment by the reviewer, we have added few sentence in section 3.6 as well as in section 4. Particularly, relationship between microbiome-metabolome-diet and host genetics is discussed in line [111-126]. An interplay between diet and microbiome using, a case study and GSMM, is stated lines [306-318].

Minor typos identified: line 156 suggest ‘Similar to’; line 185 ‘ suggest ‘validated’

RESPONSE: We thank reviewer for positive feedback. The suggested typos has been corrected in the revised manuscript.